

**Experimental study of forced convection heat transport in porous media**
**Nicola Pastore[(1)], Claudia Cherubini[(2)], Dimitra Rapti[(3)], Concetta I. Giasi[(1)]**
**[1]DICATECh, Department of Civil, Environmental, Building Engineering, and Chemistry,**
**Politecnico di Bari, Bari, Italy.**
**[2]Department of Physics & Earth Sciences, University of Ferrara, via Saragat 1 - 44122**
**Ferrara, Italy.**
**[3]New Energies And environment Company (NEA ) via Saragat, 1 - 44122 Ferrara, Italy.**
**Correspondence to: Nicola Pastore (nicola.pastore@poliba.it) and Claudia Cherubini**
**(chrcld@unif.it)**
**Abstract**
The knowledge of the dynamics of forced convection heat transfer in porous media is relevant in
order to optimize the efficiency of geothermal installations in aquifers.
In some applications groundwater is used directly as thermal fluid. The system uses one or several
drilling holes to pump and deliver groundwater with a heat exchange system at surface (open loop).
Other applications use vertical borehole heat exchangers without injection or extraction of
groundwater (closed loop). In both systems the convection flow dynamics in porous media play an
important role on the heat production.
The present study is aimed at extending this thematic issue through heat transport experiments and
their interpretation at laboratory scale. An experimental study to evaluate the dynamics of forced
convection heat transfer in a thermally isolated column filled with porous medium has been carried
out. The behavior of two porous media having different grain sizes and specific surfaces has been
observed. The experimental data have been compared with an analytical solution for one
dimensional heat transport for local non thermal equilibrium condition. The interpretation of the
experimental data shows that, the heterogeneity of the porous medium affects heat transport
dynamics causing a channeling effect which has consequences on thermal dispersion phenomena
and heat transfer between fluid and solid phases limiting the capacity to store or dissipate heat in the
porous medium.



## Introduction

The European Climate and Energy Framework for 2050 aims to shift from the massive use of fossil
sources to others characterized by very low emissions. Among the renewables sources, geothermal
energy is the only one which is available basically everywhere and at any time.
For this reason, in the recent years the use of groundwater as low-enthalpy geothermal resource for
heating and cooling of buildings and for agricultural and industrial processes is growing.
One of the main limits for the development of low – enthalpy geothermal systems concerns the high
cost of investment. Installation of geothermal energy systems requires high upfront capital
investments that often exceed the expectations of depreciation expense, so the investment is
therefore inconvenient and the economic benefits can only occur after a long time. It is therefore of
extreme importance to further the understanding of the behaviour of hydrological systems as
concerns heat transport. Studying heat transfer phenomena takes the advantage of the fact that the
governing partial differential equations used to describe flow and transport processes in porous
media are based on the same form of mass and/or energy conservation laws.
Several studies have been already carried out in this context with the aim of enhancing heat transfer
phenomena in porous media for engineering processes. Theoretical and numerical research on
convection heat transfer in porous media has used two different models for the energy equation: the
local thermal equilibrium model and the local thermal non-equilibrium model.
Most of the studies have been focussed on investigating on the validity of the local thermal
equilibrium assumption (LTE) between the solid and fluid phase, the influence of nonlinear flow
patterns, and the existing relationship between thermal dispersion and flow velocity.
Koh and Colony (1974) carried out an analytical investigation of the performance of a heat
exchanger containing a conductive porous medium using Darcy flow model, while Koh and Stevens
(1975) performed an experimental study of the same problem. They have shown that for a constant
heat flux boundary condition the wall temperature is significantly decreased by using a porous
material in the channel.
Vafai and Tien (1981) have formulated a general mathematical model that takes into consideration
the boundary and inertial (non Darcian) effects on flow and heat transfer in porous media. In
analyzing these effects, they considered three flow resistances: the bulk damping resistance due to
the porous structure, the viscous resistance due to the boundary, and the resistance due to the
inertial forces.
Later, Vafai and Tien (1982) performed a numerical and experimental investigation of the effects of
the presence of a solid boundary and inertial forces on mass transfer in porous media.



Kaviany (1985) studied laminar flow through a porous channel bounded by two parallel plates
maintained at a constant and equal temperature by applying a modified Darcy model for transport of
momentum.
Vafai and Kim (1989) considered fully developed forced convection in a porous channel bounded
by parallel plates by applying Brinkman-Forchheimer-extended Darcy model to obtain a closed-
form analytical solution.
Lauriat and Vafai (1991) presented a comprehensive review on flow and heat transfer through
porous media for two basic geometries: flow over a flat plate embedded in a porous medium and
flow through a channel filled with a porous medium.
Hadim (1994) carried out a numerical study to analyze steady laminar forced convection in a 1)
fully porous and 2) partially porous channel filled with a fluid-saturated porous medium and
containing discrete heat sources on the bottom wall.
He modelled the flow in the porous medium using the Brinkman-Forchheimer extended Darcy
model.
Kamiuto and Saitoh (1994) examined theoretically the effects of several system parameters on the
heat transfer characteristics of fully developed forced convection flow in a cylindrical packed bed
with constant wall temperatures. They developed a two-dimensional model incorporating the effects
of non-Darcy, variable porosity and radial thermal dispersion.
Hwang et al. (1995) performed a study of non-Darcian forced convection in an asymmetric heating
sintered porous channel to investigate the feasibility of using this channel as a heat sink. The study
showed that the particle Reynolds number significantly affected the solid-to-fluid heat transfer
coefficients.
A review of literature indicates that the local thermal equilibrium assumption (LTE) between the
solid and fluid phase is used in the majority of heat transfer applications involving porous media
Mikowycz et al., (1999) proposed a modified energy equation that can be solved for very early
departures from LTE conditions. Their results confirmed that local thermal equilibrium in a
fluidized bed depends on the size of the layer, mean pore size, interstitial heat transfer coefficient,
and thermophysical properties. They concluded that for a porous medium subject to rapid transient
heating, the existence of the local thermal equilibrium depends on the magnitude of a dimensionless
quantity (which they called the Sparrow number) containing the contributions of the flow in porous
media, interstitial heat transfer, and general thermal conduction.
An in-depth analysis of non-thermal equilibrium is provided by Amiri and Vafai (1994, 1998).
Amiri and Vafai (1994) carried out a steady-state analysis of incompressible flow through a bed of
uniform solid sphere particles packed randomly. The investigation was aimed at exploring the





influence of a variety of phenomena such as the inertial effects, boundary effects, and the effect of
the porosity variation model together with the thermal dispersion effect on the momentum and
energy transport in a confined porous bed. They also proved the validity of LTE assumption and the
two-dimensionality effects on transport processes in porous media.
In a subsequent study, Amiri  and Vafai (1998) realised a rigorous and flexible model to explore the
heat transfer aspects in a packed bed made of randomly oriented spherical particles. Along with the
generalized momentum equation they used a two-energy equation model to describe the thermal
response of a packed bed. They explored the temporal impact of the non Darcian terms and the
thermal dispersion effects on energy transport. In addition, they investigated on the LTE condition
and the one dimensional approach under transient condition by formulating dimensionless variables
that will serve as instruments in depicting the pertinent characteristics of energy transport in a
packed bed.
Khalil et al. (2000) performed a numerical investigation of forced convection heat transfer through
a packed pipe heated at the surface under constant heat flux showing the effects of particle
Reynolds number, pipe-to-particle diameter ratios and Prandtl number. They showed that the
average Nusselt number increases with both particle Reynolds number and Prandtl number. They
concluded that packing pipes with a porous medium can provide heat transfer enhancement for the
same pumping power.
Wu and Hwang (1998) investigated experimentally and theoretically flow and heat transfer
dynamics inside an artificial porous matrix by using a modified version of the local thermal
nonequilibrium model (LTNE) which neglected the effects of thermal dispersion in both fluid and
solid. The results showed a highly non-Fourier behaviour which combined rapid thermal
breakthrough with extremely long-tailing, that was attributed to disequilibrium between the fluid
and the porous matrix. However, the adopted model was unable to fully capture the thermal
breakthrough observed in some experimental runs. They concluded that heat transfer coefficient
increases with the decrease in porosity and the increase in the particle Reynolds number.
Emmanuel and Berkowitz (2007) were able to successfully fit the thermal breakthrough curves
obtained by Wu and Hwang (1998) by applying the continuous time random walk (CTRW) which
provided an alternative description of heat transport in porous media. They argued that larger scale
spatial heterogeneities in porous media present obstacles to both the equilibrium and the LTNE
models and that CTRW would be particularly applicable to the quantification of heat transfer in
naturally heterogeneous geological systems, such as soils and geothermal reservoirs.
Geological media are typically characterized by heterogeneities on many scales, resulting in a wide
range of fluid velocities, porosities, and effective thermal conductivities.



Despite the uncertainty and contradiction in defining the thermal dispersion, several studies
addressed the effects of thermal dispersion in porous media and different approaches have been
developed to describe it (Hsu and Cheng, 1990; Anderson, 2005; Molina-Giraldo et al., 2011).
Thermal dispersion is generally defined as a function of fluid velocity and grain size (Lu et al.,
2009, Sauty et al., 1982, Nield and Bejan, 2006).
According to Sauty et al. (1982) and Molina-Giraldo et al. (2011), the thermal dispersion is a linear
function of flow velocity and relates to the anisotropy of the velocity field whereas Rau et al. (2012)
proposed a dispersion model as a function of the square of the thermal front velocity.
The literature also contains conflicting theories about the magnitude of thermal dispersivity. Smith
and Chapman (1983) state that it has the same order of magnitude as solute dispersivities, while
Ingebritsen and Sanford (1999) neglect it. According to Vandenbohede et al. (2009) thermal
dispersivities are small in comparison to solute dispersivities and less scale-dependent.
Mori et al. (2005) showed experimentally that, for water fluxes ranging between $0.6 \times 10^{-6}$ and
$0.3 \times 10^{-3}$ (m/s) thermal dispersion was nearly independent of water flow and its effects were
insignificant.
According to Rau et al., (2012), the effect of thermal dispersion on heat transport is significant for
high values of thermal Peclet number. Also Metzger et al. (2004) introduced a dispersion model
based on the thermal Peclet number.
Koch et al. (1989) obtained an analytical expression for the dispersion tensor for a regular
arrangement of cylinders or spheres. They found that for high values of Peclet numbers, the ratio of
longitudinal total thermal diffusivity to the fluid thermal diffusivity was proportional to the square
of the Peclet number while maintaining the transverse dispersion constant. The analytic finding was
in good concordance with the experimental measurements of Gunn and Pryce (1969).
Eidsath et al. (1983) quantified the longitudinal thermal dispersion and stressed that the streamwise
ratio of longitudinal total thermal diffusivity to the fluid thermal diffusivity was proportional to
$Pe^{1.7}$.
Ait Saada et al. (2006) investigated the behaviour of microscopic inertia and thermal dispersion in a
porous medium with a periodic structure by using a local approach at the pore scale to evaluate the
velocity and temperature fields as well as their intrinsic velocity and temperature fluctuations in a
typical unit cell of the porous medium under study. They concluded that non-linear effects
characterizing microscopic inertia might be the definitive cause of thermal dispersion depending on
the nature of the porous medium and in certain situations can exceed 50% toward the contribution
of thermal dispersion. Particularly for a highly conducting fluid moving with high Peclet numbers,



microscopic inertial effects showed to take a great part in the heat transfer duty. They concluded
that a considerable interaction between the velocity and thermal fields exists.
This work is aimed at studying the dynamics of forced convection heat transport in porous media
allowing the understanding of how the grain size and the specific surface affect heat transport in
terms of macrodispersion phenomena, heat transfer between solid and fluid phases and heat storage
properties. In particular, the present study involves the experimental investigation of heat transport
through a thermally isolated column filled with porous medium. Several heat tracer tests have been
carried out using porous media with different grain sizes. The experimental observed breakthrough
curves have been compared with the one dimensional analytical solution for the forced convection
heat transport in local thermal non equilibrium condition. The results highlight the effects of grain
size and the specific surface on forced convection heat transport dynamics in porous media.
**Theoretical background**
In several studies examining the flow dynamics through porous media it is assumed that flow is
described by Darcy's law, which expresses a linear relationship between pressure gradient and flow
rate. Darcy's law has been demonstrated to be valid at low flow regimes (Re<1), whereas for
Re>>1 a nonlinear flow behavior is likely to occur. As velocity increases, the inertial effects start
dominating the flow field. In order to take these inertial effects into account, Forchheimer (1901)
introduced an inertial term representing the kinetic energy of the fluid to the Darcy equation. The
Forchheimer equation for one dimensional flow in terms of hydraulic head $h$ (L) is given as follows:
$$-\frac{dh}{dx} = \frac{\mu}{\rho g k} q + \frac{\beta}{g} q^2 \tag{1}$$
Where $x$ (L), $k$ (L$^2$) is the permeability, $\mu$ (ML$^{-1}$T$^{-1}$) is the viscosity, $\rho$ (ML$^{-3}$) is the density, $q$ (LT$^{-1}$)
is the darcy velocity and $\beta$ (L$^{-1}$) is called the non –Darcy coefficient.
Ergun (1952) derived a model for high velocity pressure loss in a porous medium from the
Forchheimer equation by correlating the permeability and inertial resistance dimensionally to the
porosity and the equivalent sphere diameter of rough particles. The permeability and inertial
coefficient are interpreted in terms of spatial parameters as follows:
$$k = \frac{d_p^{\,2} n^3}{A(1-n)^2} \tag{2}$$





$$\beta = \frac{B(1-n)}{d_p n^3} \qquad\qquad (3)$$
Where $d_p$ (L) is the average particle diameter, $n$ (-) is the porosity and the coefficients $A = 180$ and
$B = 1.8$ are empirical values and were derived by averaging the Navier – Stokes equations for a
cubic representative unit volume.
The behavior of convective heat transport in porous media is strongly dependent on the fluid
velocity and the kinetics of heat transfer process between fluid and solid phases.
Given a packed bed, within a thermally isolated column of length $L$ (L) in which a fluid flows with
a specific flow rate $q$ (LT$^{-1}$) and then with an average fluid velocity $q/n$, the initial temperature in
the column is $T_0$ (K) and a continuous flow injection transports heat energy along the column. For a
small ratio of column diameter $D$ (L) to the length $L$ and large fluid velocity the radial heat transport
dynamics can be neglected in comparison with the axial dynamics. Then the heat transport
dynamics in the porous medium column can be represented by a one dimensional model.
If the solid and fluid phases are in contact for a sufficient period of time, there is the possibility to
establish a local thermal equilibrium (LTE) condition. In such case, only one energy equation is
sufficient for the description of the convective heat transport through the porous medium. Assuming
that porosity, densities and heat capacities are constant in time the energy equation for the fluid and
solid phases are combined into a single equation as:
$$(\rho c)_{sf} \frac{\partial T_f}{\partial t} = \frac{\partial}{\partial x} \cdot \left[ -v \rho_f c_f T_f + k_{sf} \frac{\partial T_f}{\partial x} \right] \qquad\qquad (4)$$
With:
$$(\rho c)_{sf} = (1-n)\rho_s c_s + n\rho_f c_f \qquad\qquad (5)$$
$$k_{sf} = (1-n)k_s + nk_f \qquad\qquad (6)$$
Where $T_f$ (K) is the temperature of the fluid, $\rho_f$ (ML$^{-3}$) is the density of the fluid, $\rho_s$ (ML$^{-3}$) is the
density of the solid, $c_f$ (LT$^2$K$^{-1}$) is the thermal capacitance of the fluid, $c_s$ (LT$^2$K$^{-1}$) is the thermal
capacitance of the solid, $k_f$ (MLT$^{-3}$K$^{-1}$) is the thermal conductivity of the fluid, $k_s$ (MLT$^{-3}$K$^{-1}$) is the
thermal conductivity of the solid, whereas $(\rho c)_{sf}$ and $k_{sf}$ represent the equivalent heat capacity and


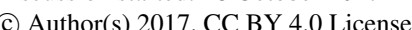

thermal conductivity of the porous domain respectively including porosity and thermal properties of
solid and fluid.
If the interaction between solid and fluid phase is rapid the solid and fluid phase cannot exchange
sufficient amount of energy to establish local thermal equilibrium. At a given location solid and
fluid phases have different temperatures. In the local thermal non equilibrium (LTNE) condition
each phase needs an energy equation for the description of heat transport. Assuming that porosity,
densities and heat capacities are constant in time, the energy equations can be written for the fluid
and solid phase:
$$n\rho_f c_f \frac{\partial T_f}{\partial t} = \frac{\partial}{\partial x} \cdot \left[ -vn\rho_f c_f T_f + nk_f \frac{\partial T_f}{\partial x} \right] + q_{fs} \qquad (7)$$

$$(1-n)\rho_s c_s \frac{\partial T_s}{\partial t} = \frac{\partial}{\partial x} \cdot \left[ (1-n)k_s \frac{\partial T_s}{\partial x} \right] - q_{fs} \qquad (8)$$

The interaction between the two phases is represented by the sink/source terms $q_{fs}$ given by
following equation:
$$q_{fs} = hs_f \left( T_s - T_f \right) \qquad (9)$$

Where $h$ (MT$^{-3}$K$^{-1}$) is the convective heat transfer coefficient and $s_f$ (L$^{-1}$) is the specific surface area.
The convective heat transfer coefficient is related to the Nusselt number Nu that for the porous
medium can be expressed as:
$$\text{Nu} = \frac{q_{fs}d_p}{k_f \left( T_f - T_s \right)} = \frac{hd_p}{k_f} \qquad (10)$$

The hydrodynamic mixing of the interstitial fluid at the pore scale gives rise to significant thermal
dispersion phenomena. Generally, the hydrodynamic mixing is due to the presence of obstruction,
flow restriction and turbulent flow. Therefore, the equivalent thermal conductivity in equation (1)
and thermal conductivity in equation (4) is replaced with the effective thermal conductivity $k_{eff}$
which is the sum of thermal conductivity and thermal dispersion conductivity. The effective thermal
conductivity depends on various parameters such as mass flow rate, porosity, shape of pores,
temperature gradient, and solid and fluid thermal properties (Kaviany, 1995). The following
equation can be used to estimate $k_{eff}$.





$$\frac{k_{eff}}{k_f} = \frac{k}{k_f} + K \cdot \text{Pe}^a \qquad (11)$$
Pe represents the Peclet number defined as the product between the Reynolds number Re and
Prandtl number Pr;
$$\text{Pe} = \text{Re} \times \text{Pr} = \frac{\rho_f v d_p}{\mu} \times \frac{c_f \mu}{k_f} = \frac{v d_p}{D_f} \qquad (12)$$
The energy equation representative of the local thermal non equilibrium can be written as:
$$\frac{\partial T_f}{\partial t} = -v \frac{\partial T_f}{\partial x} + D_{eff} \frac{\partial^2 T_f}{\partial x} + \alpha \left( T_s - T_f \right) \qquad (13)$$
$$\frac{1-n}{n} \frac{\rho_s c_s}{\rho_f c_f} \frac{\partial T_s}{\partial t} = \frac{1-n}{n} \frac{k_s}{\rho_f c_f} \frac{\partial^2 T_s}{\partial x} - \alpha \left( T_s - T_f \right) \qquad (14)$$
With:
$$D_{eff} = \frac{k_{eff}}{\rho_f C_f} \qquad (15)$$
$$\alpha = \frac{h s_f}{\rho_f C_f} \qquad (16)$$
$D_{eff}$ ($L^2 T^{-1}$) is the thermal dispersion and $\alpha$ ($T^{-1}$) is the exchange coefficient.
The thermal dispersion happens due to hydrodynamic mixing of fluid at the pore scale caused by the
nature of the porous medium. Greenkorn (1983) found nine mechanisms responsible of most of the
mixing among which the following: 1) Mixing caused by the tortuosity of the flow channels due to
obstructions: fluid elements starting a given distance from each other and proceeding at the same
velocity will not remain the same distance apart; 2) Existence of autocorrelation in flow paths: in this
case, all pores in a porous medium may not be accessible to a fluid element after it has entered a
particular flow path; 3) Recirculation due to local regions of reduced pressure due to the conversion of
pressure energy into kinetic energy; 4) Hydrodynamic dispersion in a capillary caused by the velocity
profile produced by the adhering of the fluid to the wall; 5) Molecular diffusion into dead-end pores: as
solute rich front passes the pore. After the front passes, the solute will diffuse back out and thus,
dispersing.




Using the analogy with the solute transport the Damköhler number Da (Leij et al., 2012) can be
introduced in order to evaluate the influence of heat transfer between the fluid and solid phases on
the convection phenomena:
$$\text{Da} = \frac{\alpha L}{v} \qquad (17)$$
When Da reaches the unit the heat transfer time scale is comparable with the convection time scale
and the LTNE exists between solid and fluid phases. At very high values of Da the heat transfer
time scale is much lower than convective time scale and the LTE condition exists between solid and
fluid phases. Finally, at very low values of Da the heat transfer phenomena can be neglected.
Neglecting the first term on the right side of the Equation 14, the analytical solution of the system
equations describing 1D heat transport in semi – infinite domain for instantaneous temperature
injection is given by Goltz and Robertz (1986). According to this analytical solution, the probability
of density function $PDF_{LTNE}$ of the residence time for LTNE condition can be written as:
$$PDF_{LTNE}(x,t) = e^{\alpha t} c_0(x,t) + \alpha \int_0^t H(t,\tau) c_0(x,t) d\tau \qquad (18)$$
With:
$$c_0(x,t) = \frac{1}{\sqrt{\pi D_{eff} t}} \exp\left(\frac{x - vt}{4 D_{eff} t}\right) \qquad (19)$$
$$H(t,\tau) = e^{-\frac{\alpha}{\beta}(t-\tau) - \alpha \tau} \frac{\tau I_1\left(\frac{2\alpha}{\beta}\sqrt{\beta(t-\tau)\tau}\right)}{\sqrt{\beta(t-\tau)\tau}} \qquad (20)$$
$$\beta = \frac{1-n}{n} \frac{\rho_s c_s}{\rho_f c_f} \qquad (21)$$
Where $I_1$ is the modified Bessel function of order 1.
The coefficient $\alpha$ can be viewed as the reciprocal of the exchange time required to transfer energy
from fluid to solid phase and vice versa. The parameter $\beta$ (-) represents the ratio between the
volume specific heat capacity of the solid phase and the fluid. The effect of local thermal non
equilibrium is stronger when the exchange time is the same order of magnitude of the transport
time. The local thermal non equilibrium is characterized by thermal distribution profile with a
tailing effect.



The observed temperature function $T_{obs}(t)$ at a generic distance $x$ from the injection temperature
function $T_{inj}(t)$ can be obtained using the convolution theorem:
$$T_{obs}(x,t) = T_{inj}(0,t) * PDF_{LTNE}(x,t) \tag{22}$$

**Experimental setup**

The test on convective heat transport in the porous medium has been conducted on a laboratory
physical model. Figure 1 shows a sketch of the experimental setup. A plastic circular pipe
characterized by a diameter of $D = 0.11$ m and height of $H = 1.66$ m has been thermally insulated
using a roll of elastomeric foam with a thickness of $s = 0.04$ m and a thermal conductivity of $\lambda =$
$0.037$ Wm$^{-1}$K$^{-1}$. The pipe can be filled with different porous materials with different grain sizes and
hydrothermal properties. Seven thermocouples have been equally placed along the axis of the pipe
with a reciprocal distance of 0.185 m. The first thermocouple is located at a distance of 0.435 m
from the inlet of the water. TC08 Thermocouple Data Logger (pico Thecnology) with sampling rate
equal to 1 second has been connected with the thermocouples. An adaptable constant head reservoir
and an outlet reservoir permit to maintain a constant head during the test and water within the pipe
flows from the bottom to the top. An ultrasonic velocimeter (DOP3000 by Signal Processing) is
used to measure the instantaneous flow rate. An electric water boiler characterized by a volume
equal to 0.01 m$^3$ has been used to heat the water flowing through the pipe.
A medium gravel (M$_1$) (USDA, 1975) and a very coarse gravel (M$_2$) (USDA, 1975) have been
used.. The Figure 2 shows the tested materials whereas in Table 1 are reported the hydraulic and the
thermal parameters of each material. The grain size and the specific surface of each porous material
is directly estimated on a sample of one hundred grains. Whereas the porosity is estimated by the
ratio between the volume void space and the total volume of the filled plastic circular pipe. The
volume of the void space is obtained measuring the amount of water which enters in the pipe until
full saturation. The thermal characteristics reported in the table 1 are literature values
(www.engineeringtoolbox.com).   The temperature tracer tests involve the observation of the
thermal breakthrough curves (BTCs) monitored by the seven thermocouples. Initially cold water
flows through the pipe filled with the porous medium in order to have a constant temperature $T_0$
along the pipe. Subsequently hot water is flows through the pipe, maintaining a constant head
condition during the test.

**Discussion**



For each tested porous medium four thermal tracer tests have been carried out varying Re in the
range 5.7– 22.5 for $M_1$ and 23.5 – 105.5 for $M_2$. The thermal BTCs observed at different distances
have been fitted together using equation (19). The root mean square error (RMSE) and the
determination coefficient ($r^2$) have been used as criteria to evaluate the goodness of the fitting. The
parameters $v$, $D_{eff}$ and $\alpha$ have been individually fitted for each thermal tracer test whereas $\beta$ has
been imposed constant for all tracer tests of each tested porous medium. Table 2 shows the
estimated values of the heat transport parameters, the RMSE and $r^2$, whereas figure 3 and figure 4
show the fittings results of the observed temperature distribution along the porous column for $M_1$
and $M_2$ respectively. Table 4 shows the dimensionless numbers Pe, $k_{eff}/k_f$, Nu and Da evaluated for
the different values of Re.
As shown in Table 2, the fluid velocity $q/n$ is systematically higher than the estimated thermal
convective velocity $v$ for the medium gravel $M_1$, contrarily for the very coarse gravel $M_2$ $q/n$ is
systematically lower than v.
This phenomenon for the coarser material might be attributable to the fact that the heat propagates
through both the solid and fluid phase (Anderson, 2005, Rau et al. 2012) and the existence of
channeling phenomena that might also have an influence in increasing the convective heat.
Even for finer grained materials (2 mm), Rau et al (2012) also found values of thermal velocity
systematically lower than solute velocity, coherently with Bodvarssoon (1972), Oldenburg and
Pruess (1998), Geiger et al. (2006).
Another discrepancy has been observed comparing the values of the porosity presented in table 1
and the value of porosity obtained from the equation 18 equal to 0.467 and 0.469 respectively for
$M_1$ and $M_2$. For $M_1$ the value of porosity presented in table 1 reaches the value derived from $\beta$.
Whereas for $M_2$ the value presented in table 1 is higher than the value derived from $\beta$.
These results highlight that for $M_2$ there is the existence of stagnant zones which reduce the amount
of porosity that contributes to fluid flow. In other words, in $M_1$ the total porosity reaches to the
effective porosity, whereas in $M_2$ the effective porosity is less than total porosity.
In figure 5 is reported the relationships between Pe and the ratio between the effective thermal
conductivity and the fluid thermal conductivity $k_{eff}/k_f$ . The experimental results show a non linear
behavior well represented by equation (8). A change of slope is evident changing from $M_1$ to $M_2$.
The latter material shows a more pronounced thermal dispersion caused by the hydrodynamic
mixing of fluid at the pore scale. Some mixing is caused by the tortuosity of the flow paths due to the



presence of obstructions: the fluid elements starting a given distance from each other and
proceeding at the same velocity will not remain at the same distance apart. The high level of flow
path heterogeneity gives rise to a higher velocity variation at pore scale as well as the presence of
the preferential flow paths that enhance the effect of macrodispersion. Mixing can also be caused by
recirculation caused by local regions of reduced pressure arising from flow restrictions.
Further mixing can arise from the fact that all pores in a porous medium may not be accessible to a
fluid element after it has entered a particular flow path.
These results are coherent with those obtained by Rau et al (2012) who found that the thermal
dispersion was transitioning between not depending on the flow velocity and a non linear increase
with velocity. They affirmed that the location of the transition zone is a function of the thermal
properties of the solid and the sedimentological architecture.
Figure 6 shows the relationship between Pe and Nu. The experimental results highlight that the
Nusselt number can be represented by an equation like $Nu = C \times Pe$, where $C$ (-) is a coefficient that
assumes a value equal to 0.41 for $M_1$ and 0.03 for $M_2$. This coefficient has the physical meaning of
the ratio between the surface of the grains in contact with the active flow path that transports heat
and the total surface of the grain. $M_2$ respect to $M_1$ is characterized by the presence of preferential
flow paths and then an equal number of Pe corresponding to a lower Nu because the surface of the
grain available to exchange heat between the fluid and solid phase is lower.
As shown in table 3 the Damköhler number Da calculated for $M_1$ is greater than the unit. Heat
exchange is so rapid giving rise to an instantaneous equilibrium between solid and fluid phase. The
heat has enough time to diffuse in solid phase. Contrarily Da calculated for $M_2$ is close to the unit,
there is the presence of the local thermal non equilibrium condition.
A comparison of the heat $J$ ($ML^2T^{-2}$) stored in the porous column per unit temperature difference
$\Delta T = T_{inj} - T_0$ (K) varying the specific discharge $q$ for each tested porous medium can be evaluated
considering a continuous temperature injection function as:

$$\frac{J}{\Delta T} = \rho_f c_f Q \int_0^\infty \left(1 - \int_0^t PDF(L,\tau)d\tau\right)dt \qquad (23)$$

The combined effects of the flow rate and the particle diameter on heat transfer are illustrated in
Figure 7 that shows the variation of the heat stored in the column per unit temperature difference
varying the specific discharge for $M_1$ and $M_2$. As the flow rate increases, the stored heat increases,
and the porous medium with a smaller particle diameter generates a higher increase in heat transfer
enhancement than one with a larger particle diameter. This is coherent with the results obtained by
Dehghan and Aliparast (2011) and Kifah (2004). $M_1$ permits to store more heat than $M_2$. The former





is characterized by a more homogeneous flow path distribution that allows a greater interaction
between fluid and solid phase. On the contrary $M_2$ has a more heterogeneous flow path distribution
that increases thermal macrodispersion phenomena at the pore scale and at the same time reduces
the interaction between the fluid and solid phase.
In order to put into evidence the performance of the heat transfer enhancement of the porous
materials it can be useful to compare the Nusselt number and the hydraulic head loss $dh/dx$
evaluated by equation 1. The Figure 8 shows the ratio between the Nusselt number and head loss as
function of the Peclet number. Despite $M_1$ presents a higher heat transfer enhancement respect to
$M_2$, the head losses are higher and then the ratio between Nu and $dh/dx$ is lower. Furthermore, as Pe
increases, the heat transfer enhancement increases more rapidly than the head loss for the fine
material $M_1$, whereas for the coarser material $M_2$ the opposite happens: increasing the Peclet
number the Nusselt number increases weakly due to the presence of channeling phenomena that
reduce the heat exchange area between fluid and solid phases.

**Conclusion**
In this study a laboratory physical model has been set up to analyze the behavior of forced
convective heat transport in two porous media characterized by different grain sizes and specific
surfaces. For each material four tracer tests have been carried out and they have been compared
with the 1D analytical solution of LTNE model. The flow paths heterogeneity that characterizes the
coarser material gives rise to a higher velocity variation at pore scale with a channeling effect which
causes: 1) the increase in the macrodispersion phenomena in the forced convection heat transport,
2) the decrease in the surface of the grain available to exchange heat between the fluid and solid
phase, 3) the presence of the local thermal non equilibrium condition 4) the decrease in the amount
of heat that can be stored in the porous medium and 5) a weak growth of heat transfer enhancement
respect to the head loss as convective phenomena increases.
The finer material $M_1$ has a more homogeneous flow path distribution that allows a greater
interaction between fluid and solid phase and therefore allows to store more heat than the coarser
one.
This can also be seen analyzing the ratio between the Nusselt number and the head loss as function
of Peclet number for both materials. Even though the coarser material $M_2$ is more permeable than
$M_1$ as the advective phenomena increase, the head loss increases more rapidly than the heat transfer





enhancement due to the channeling effect that increases the macrodispersion phenomena and
reduces the heat transfer between fluid and solid phase.
The experimental results emphasize the differences between porous and fractured media. As
observed by Cherubini at al. (2017) a fractured medium with high density of fractures and then with
a higher specific surface is not efficient to store thermal energy because the fractures are surrounded
by a matrix with a more limited capacity to store heat. An opposite behavior has been observed in
porous media in which a higher specific surface corresponds to a higher capacity to store heat. For
porous media as the specific surface decreases the macrodispersion phenomena increase due
essentially to the channeling effect and then the surface of the grain available to exchange heat
between the fluid and solid phase decreases. Whereas for the fractured media this statement is not
true because the macrodispersion phenomena are more related contrarily to the roughness and
aperture variation of each single fracture as well as to the connectivity of the fracture network.
The study has increased the understanding of heat transfer processes in the subsurface encouraging
the investigation on how further parameters such as the shape and the roughness of the grain of
porous media affect the amount of energy that can be stored. This is important to maximize the
efficiency and minimize the environmental impact of the geothermal installations in groundwater.

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





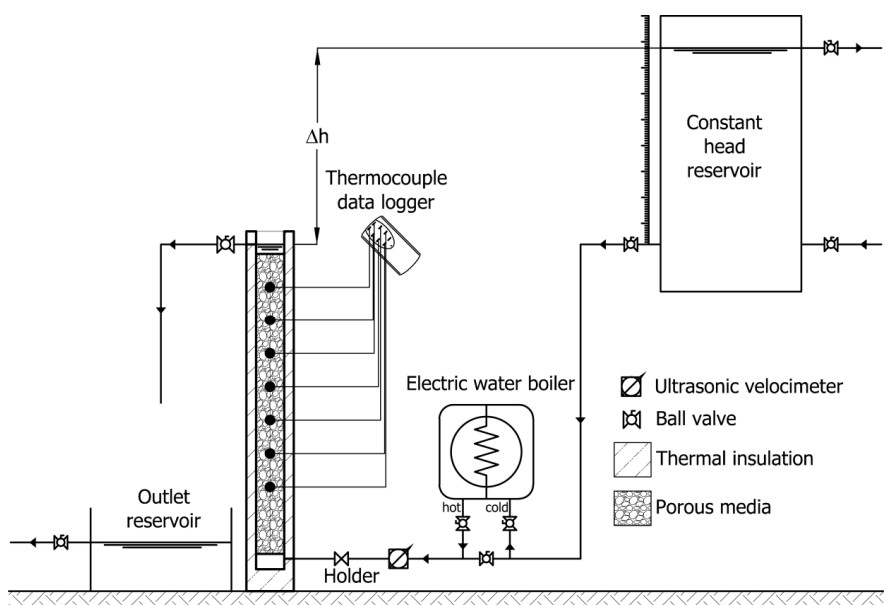


**Figure 1. Setup of experimental apparatus**

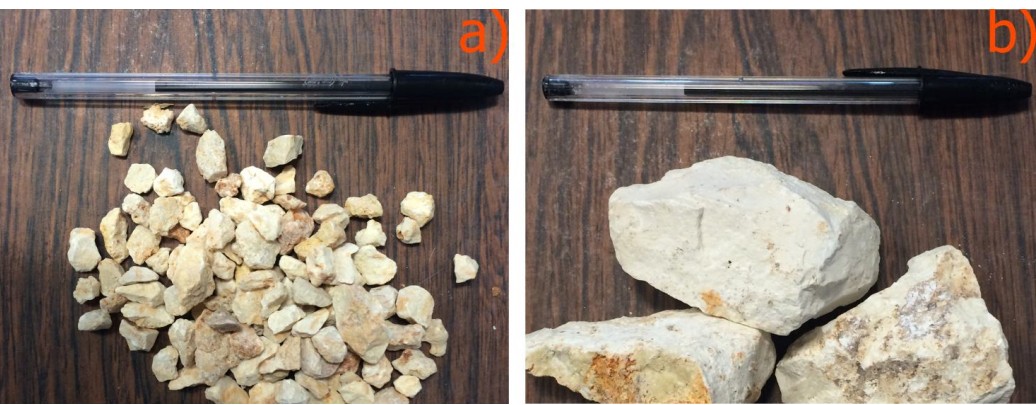


**Figure 2. Samples of the materials used for the experiments with different average grain sizes $d_p$. a) $d_p$ = 9.2 mm b) $d_p$ = 41.6 mm.**


| | $M_1$ | $M_2$ |
|---|---|---|
| Porosity (-) | 0.47 | 0.53 |
| Average grain size (mm) | 9.21 | 41.65 |
| Average specific surface (m$^{-1}$) | 675.80 | 148.4 |
| Soild density (Kg·m$^{-3}$) | 2210 | 2210 |
| Soil heat capacity (J·Kg$^{-1}$·K$^{-1}$) | 840 | 840 |
| Soil thermal conductivity (W·m$^{-1}$·K$^{-1}$) | 2.15 | 2.15 |

**Table 1. Properties of the porous materials**

| Re | $q/n \times 10^{-2}$ (m/s) | $v \times 10^{-2}$ (m/s) | $D_{eff} \times 10^{-3}$ (m$^2$/s) | $\alpha$ (s$^{-1}$) | $\beta$ (-) | RMSE | $r^2$ |
|---|---|---|---|---|---|---|---|



| | Re | | | | | | | |
|---|---|---|---|---|---|---|---|---|
| | 5.7 | 0.134 | 0.109 | 0.099 | 0.144 | | 75.659 | 0.9781 |
| **M₁** | 9.5 | 0.223 | 0.222 | 0.102 | 0.260 | | 4.835 | 0.9958 |
| | 15.5 | 0.361 | 0.321 | 0.116 | 0.403 | | 1.176 | 0.9984 |
| | 22.5 | 0.525 | 0.486 | 0.126 | 0.767 | 0.480 | 0.033 | 0.9999 |
| | 23.5 | 0.106 | 0.138 | 0.165 | 0.003 | | 29.740 | 0.9815 |
| **M₂** | 46.9 | 0.211 | 0.248 | 0.273 | 0.006 | | 7.843 | 0.9886 |
| | 69.3 | 0.312 | 0.367 | 0.409 | 0.008 | | 4.742 | 0.9904 |
| | 105.5 | 0.475 | 0.579 | 0.655 | 0.012 | 0.476 | 1.747 | 0.9944 |

**Table 2. Estimated values of parameters for LTNE model for different Re values.**

| | Re | Pe | $k_{eff}/k_f$ | Nu | Da |
|---|---|---|---|---|---|
| | 5.7 | 70.05 | 688.83 | 26.51 | 146.28 |
| **M₁** | 9.5 | 142.49 | 711.48 | 47.98 | 130.18 |
| | 15.5 | 205.95 | 811.60 | 74.30 | 139.47 |
| | 22.5 | 311.64 | 882.13 | 141.45 | 175.46 |
| | 23.5 | 402.32 | 1148.61 | 11.32 | 2.10 |
| **M₂** | 46.9 | 721.64 | 1902.80 | 25.21 | 2.61 |
| | 69.3 | 1068.78 | 2856.49 | 33.52 | 2.34 |
| | 105.5 | 1684.17 | 4568.91 | 50.84 | 2.25 |

**Table 3. Dimensionless numbers Pe, $k_{eff}/k_f$, Nu and Da calculated for different Re values.**



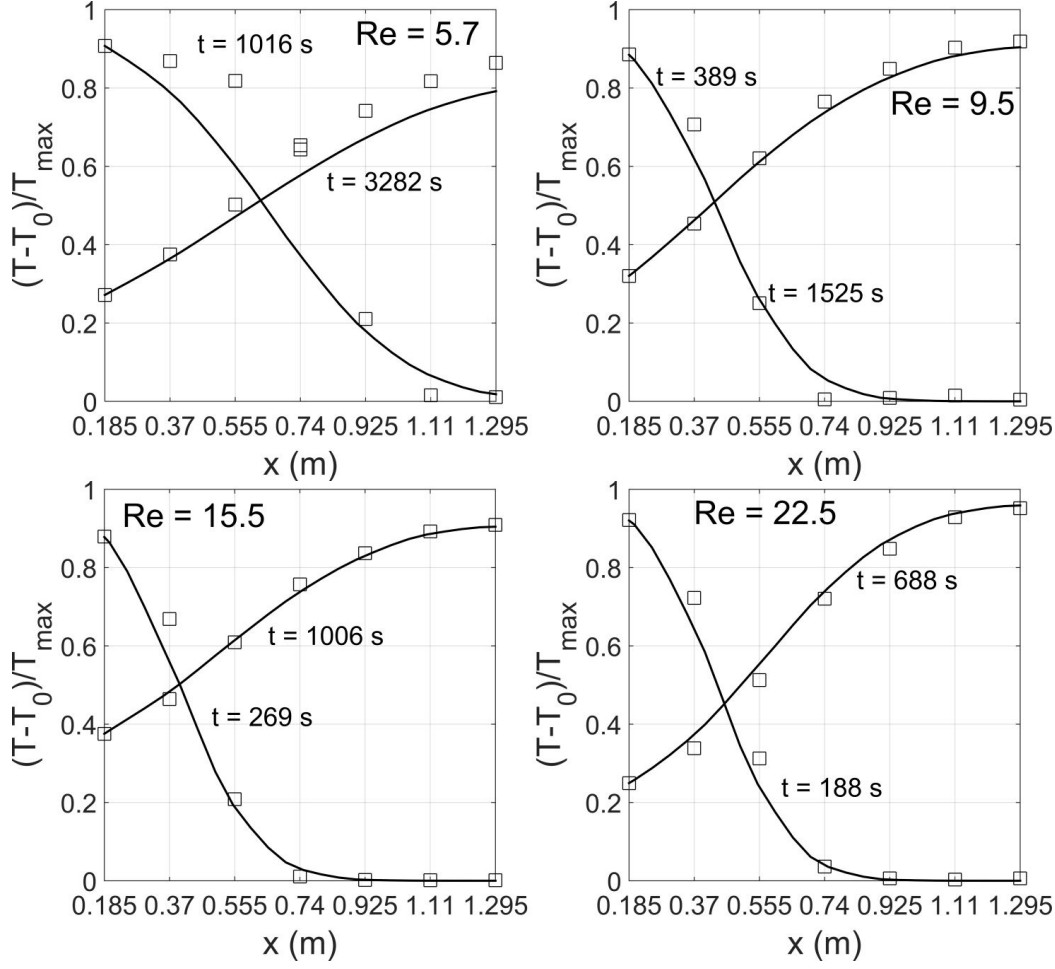


**Figure 3. Temperature distribution for different Re values along the porous column filled with material $M_1$. Squares represent the experimental values, the continuous lines represent the simulated values.**





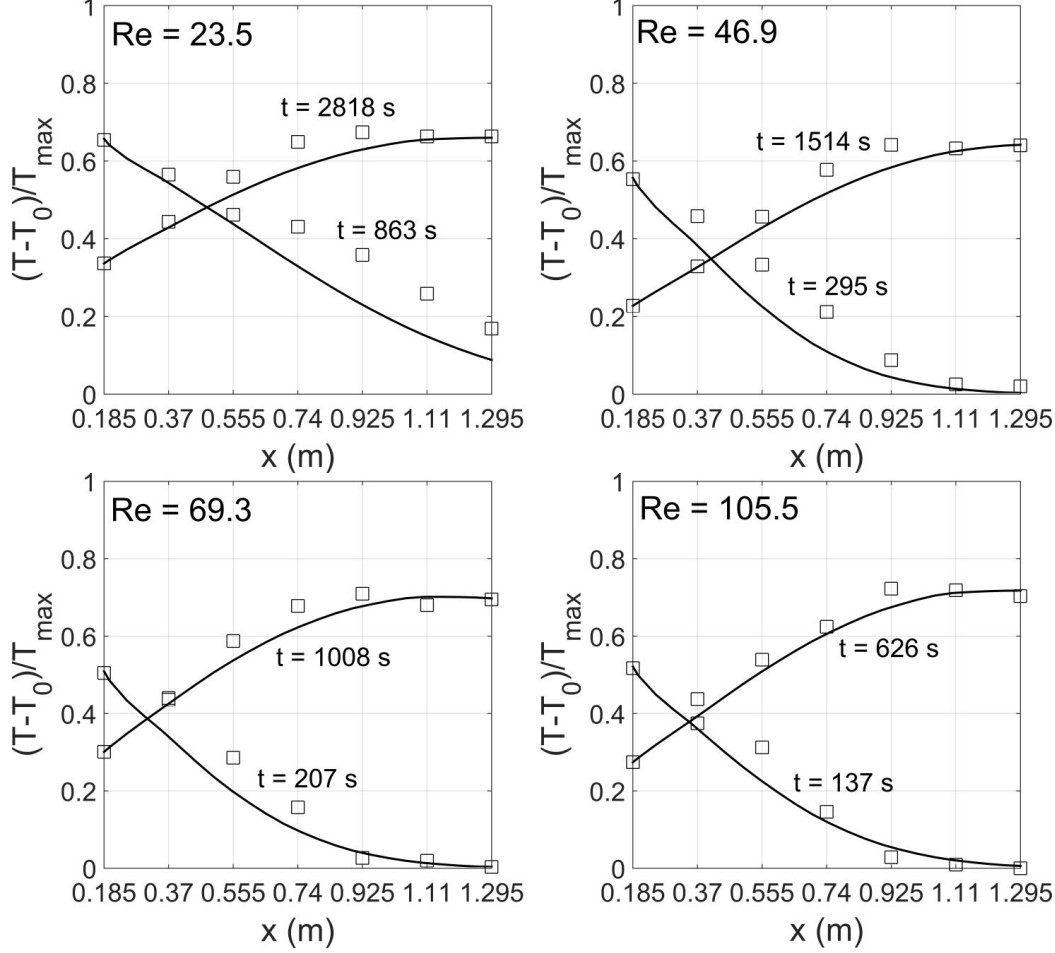


**Figure 4. Temperature distribution for different Re values along the porous column filled with material M₂. Squares represent the experimental values, continuous lines represent the simulated values.**






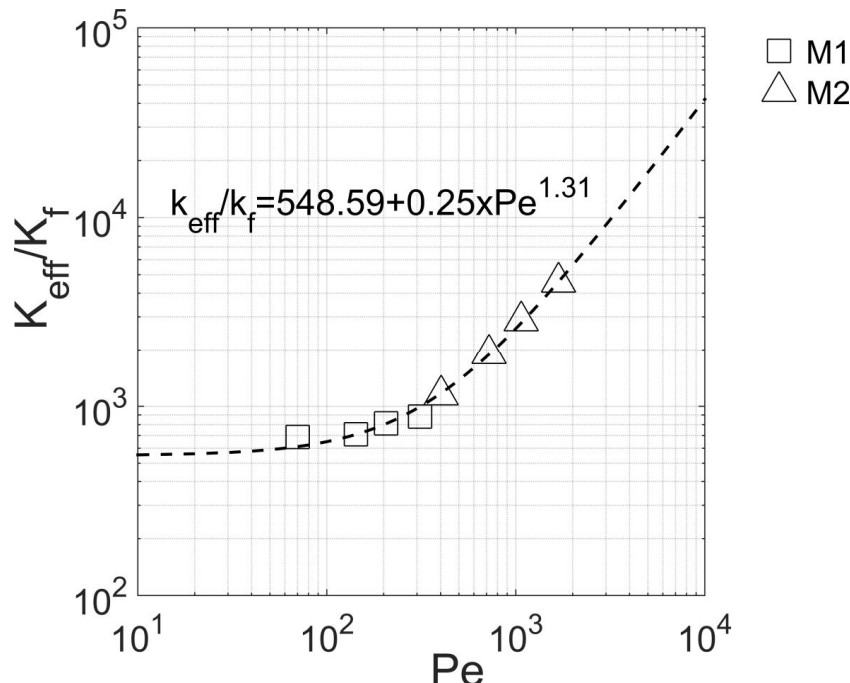


**Figure 5. Relationship between *Pe* and $k_{eff}/k_f$.**

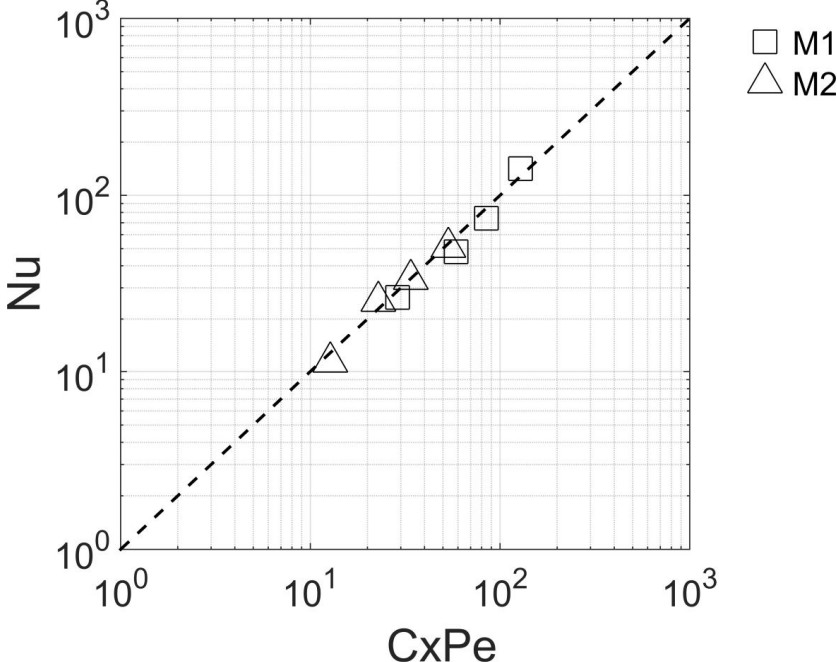


**Figure 6. Relationship between C×Pe and Nu.**



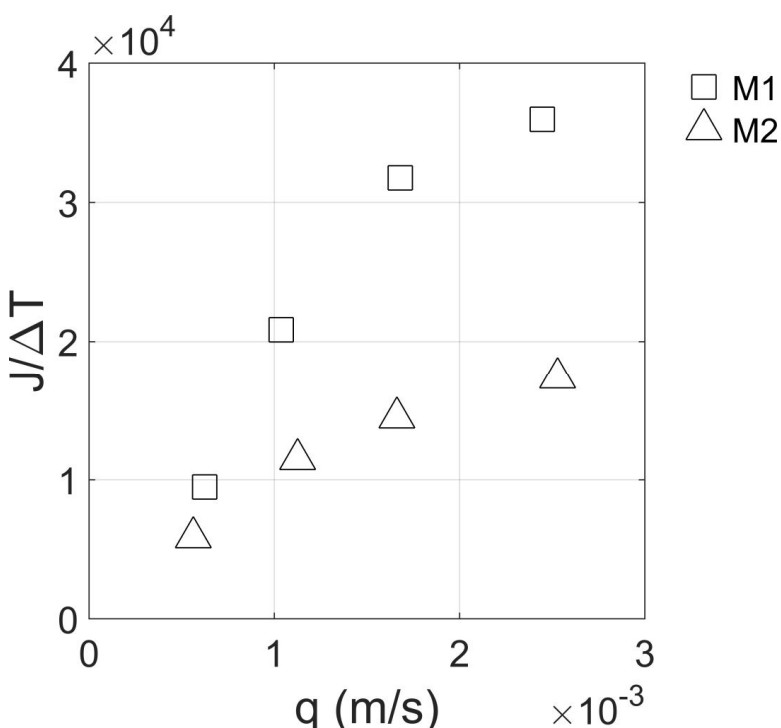


**Figure 7. heat exchanged $J/(T_{inj} - T_0)$ varying the specific rate $q$ (m/s).**

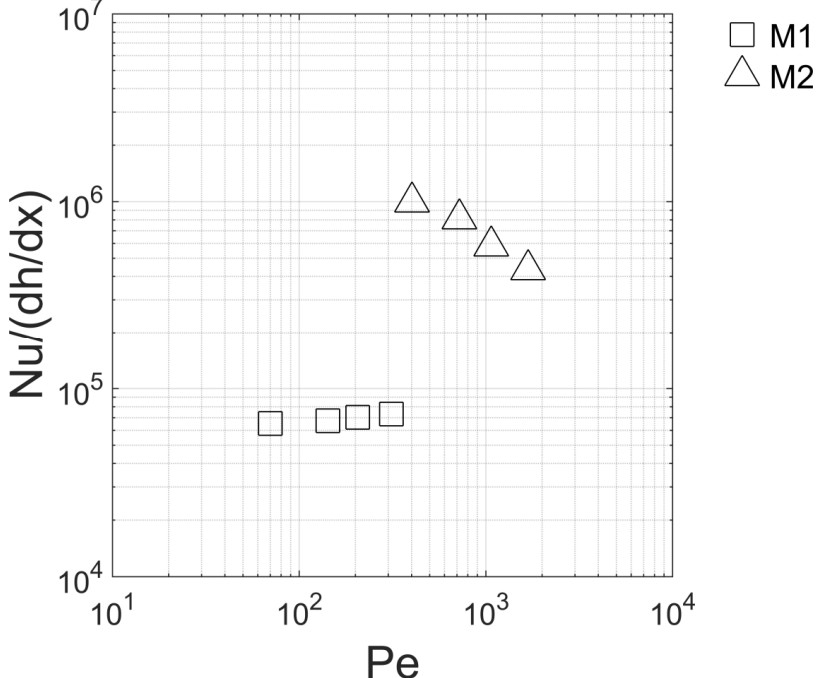





**Figure 8. Nusselt number – head loss ratio varying Peclet number.**