# Peer review of "Experimental study of forced convection heat transport in porous media"

_Nonlinear Processes in Geophysics, 2017_

## Referee Comment (RC1) · Anonymous Referee #1 · 21 Nov 2017

The following items are needed to be discussed:

1. Please check Eqs (4) and (16). The advection term should be multiplied further by the porosity in Eq (4). $\alpha$ should be divided further by the porosity in Eq (16).

2. How do the authors obtain the specific surface area in the present work? This was not introduced.

3. In Fig 3, why the inlet temperature was first higher then lower than the downstream temperature? It seems that some information is missing in the introduction of the experiment procedure.

4. To validate the present results, it is recommended to compare the results of the convectional heat transfer coefficient and effective thermal conductivity with the current

classical correlations.

5. In my opinion, the expression of v=q/n is obtained rigidly from the volume averaging theory. Thus, v should be taken as a known constant in the data processing, as well as $\beta$. Of course, the RMSE will be larger if so, but I think the experiment results are allowed to have larger errors. Please comment on this.
* * *

---

## Referee Comment (RC2) · Anonymous Referee #2 · 1 Jan 2018

Authors extended a thematic issue through heat transport experiments and their interpretation at laboratory scale. They carried out an experimental study to evaluate the dynamics of forced convection heat transfer in a thermally isolated column filled with porous medium. The behavior of two porous media having different grain sizes and specific surfaces was observed. The analysis is interesting. But I have the following concerns: (1) The abstract part should be reduced to highlight the more important contents. (2) This paper focuses on the experimental study of forced convection heat transport in porous media. So many related works have been listed in introduction part. Compared to the existing methods and results, what is the main contributions of this paper? The authors should explicitly specify this. (3) Authors claimed that for M2 there is the existence of stagnant zones which reduce the amount of porosity that

contributes to fluid flow. Add more analysis for this. (4) All the parameters should be explained. Some parts are hard to follow. (5) Please highlight the importance of this experimental study. And clearly indicate what is the new question that is answered in this experimental study? (6) Add more words for Fig.3 and 4. (7) Authors indicated that "the behavior of two porous media having different grain sizes and specific surfaces has been observed." If possible, please provide snapshoot for this. (8) What kind of experimental data was acquired? How to interpret these experimental data? All these need to be clearly indicated.

---

## Author Comment (AC1) · 6 Feb 2018

1) Please check Eqs (4) and (16). The advection term should be multiplied further by the porosity in Eq (4). $\alpha$ should be divided further by the porosity in Eq (16). The equation (4) and (16) has been corrected. 2) How do the authors obtain the specific surface area in the present work? This was not introduced. The expression to determine the specific surface area has been introduced (equation 10). 3) In Fig 3, why the inlet temperature was first higher then lower than the downstream temperature? It seems that some information is missing in the introduction of the experiment procedure. The experimental setup is realized in the way that the inlet has a non-continuous injection (10 litres), so when the injection terminates the temperature registered by the first thermocouple becomes lower than the one registered by the last thermocouple. 4) To validate the present results, it is recommended to compare the results of the convectional heat transfer coefficient and effective thermal conductivity with the current. Regarding the Nusselt number and the heat transfer coefficient, the experimental results have been compared with Wakao correlation (1979) and the experimental correlation between volumetric Nusselt number and Reynolds number found in Fu et al. (1998), Kamiut and Yee (2005), Ando et al. (2013). The relationship between the keff/kf and Pe has been compared with the experimental results found by Levec and Carbonell (1985), Gunn and Price (1969), Pfancuch (1963), Ebach and White (1958). 5) In my opinion, the expression of v=q/n is obtained rigidly from the volume averaging theory. Thus, v should be taken as a known constant in the data processing, as well as $\beta$. Of course, the RMSE will be larger if so, but I think the experiment results are allowed to have larger errors. Please comment on this We have preferred not to constrain the thermal convective velocity v to values v=q/n. Because, first of all n represents the value of total porosity and therefore the convective velocity for a conservative solute should be equal to v = q/ne where ne represents the effective porosity; second the thermal convective velocity should be less than conservative solute velocity as reported in Bodvarssoon (1972), Oldenburg and Pruess (1998), Geiger et al. (2006). The first consideration is more relevant for M2, whereas the second consideration is more relevant for M1.

Please also note the supplement to this comment:
https://www.nonlin-processes-geophys-discuss.net/npg-2017-53/npg-2017-53-AC1-supplement.pdf

**Supplement:**

[revised manuscript text omitted]

Heat transfer dynamics can be represented also by the volumetric Nusselt number $\text{Nu}_v$:

$$\text{Nu}_v = \frac{h s_f d_p^2}{k_f} \qquad (12)$$

Wakao et al. (1979) found the following correlation for the volumetric Nusselt number:

$$\text{Nu}_v = 2.0 + 1.1 \text{Re}^{0.6} \text{Pr}^{1/3} \qquad (13)$$

Fu et al. (1998), Kamiut and Yee (2005) and Ando et al. (2013) based on experimental data found a correlation between the volumetric Nusselt number and Reynolds number:

$$\text{Nu}_v = C \text{Re}^m \qquad (14)$$

The hydrodynamic mixing of the interstitial fluid at the pore scale gives rise to significant thermal dispersion phenomena. Generally, the hydrodynamic mixing is due to the presence of obstruction, flow restriction and turbulent flow. Therefore, the equivalent thermal conductivity in equation (4)

and thermal conductivity in equation (7) is replaced with the effective thermal conductivity $k_{eff}$

which is the sum of thermal conductivity and thermal dispersion conductivity. The effective thermal conductivity depends on various parameters such as mass flow rate, porosity, shape of pores, temperature gradient, and solid and fluid thermal properties (Kaviany, 1995). The following equation can be used to estimate $k_{eff}$.

$$\frac{k_{eff}}{k_f} = \frac{k}{k_f} + K \cdot \text{Pe}^a \qquad (15)$$

Pe represents the Peclet number defined as the product between the Reynolds number Re and

Prandtl number Pr;

$$\text{Pe} = \text{Re} \times \text{Pr} = \frac{\rho_f v d_p}{\mu} \times \frac{c_f \mu}{k_f} = \frac{v d_p}{D_f} \qquad (16)$$

The energy equation representative of the local thermal non equilibrium can be written as:

$$\frac{\partial T_f}{\partial t} = -v \frac{\partial T_f}{\partial x} + D_{eff} \frac{\partial^2 T_f}{\partial x} + \alpha (T_s - T_f) \qquad (17)$$

$$\frac{1-n}{n} \frac{\rho_s c_s}{\rho_f c_f} \frac{\partial T_s}{\partial t} = \frac{1-n}{n} \frac{k_s}{\rho_f c_f} \frac{\partial^2 T_s}{\partial x} - \alpha (T_s - T_f) \qquad (18)$$

With:

$$D_{eff} = \frac{k_{eff}}{\rho_f C_f} \qquad (19)$$

$$\alpha = \frac{h s_f}{n \rho_f C_f} \tag{20}$$

$D_{eff}$ ($L^2 T^{-1}$) is the thermal dispersion and $\alpha$ ($T^{-1}$) is the exchange coefficient.

The thermal dispersion happens due to hydrodynamic mixing of fluid at the pore scale caused by the
nature of the porous medium. Greenkorn (1983) found nine mechanisms responsible of most of the
mixing among which the following: 1) Mixing caused by the tortuosity of the flow channels due to
obstructions: fluid elements starting a given distance from each other and proceeding at the same
velocity will not remain the same distance apart; 2) Existence of autocorrelation in flow paths: in this
case, all pores in a porous medium may not be accessible to a fluid element after it has entered a
particular flow path; 3) Recirculation due to local regions of reduced pressure due to the conversion of
pressure energy into kinetic energy; 4) Hydrodynamic dispersion in a capillary caused by the velocity
profile produced by the adhering of the fluid to the wall; 5) Molecular diffusion into dead-end pores: as
solute rich front passes the pore. After the front passes, the solute will diffuse back out and thus,
dispersing.

Using the analogy with the solute transport the Damköhler number Da (Leij et al., 2012) can be
introduced in order to evaluate the influence of heat transfer between the fluid and solid phases on
the convection phenomena:

$$\mathrm{Da} = \frac{\alpha L}{v} \tag{21}$$

When Da reaches the unit the heat transfer time scale is comparable with the convection time scale
and the LTNE exists between solid and fluid phases. At very high values of Da the heat transfer
time scale is much lower than convective time scale and the LTE condition exists between solid and
fluid phases. Finally, at very low values of Da the heat transfer phenomena between solide and fluid
phase can be neglected.

Neglecting the first term on the right side of the Equation 18, the analytical solution of the system
equations describing 1D heat transport in semi – infinite domain for instantaneous temperature
injection is given by Goltz and Roberts (1986). According to this analytical solution, the probability
of density function $PDF_{LTNE}$ of the residence time for LTNE condition can be written as:

$$PDF_{LTNE}\left(x,t\right) = e^{\alpha t} PDF_0\left(x,t\right) + \alpha \int_0^t H\left(t,\tau\right) PDF_0\left(x,t\right) d\tau \tag{22}$$

With:

$$PDF_0\left(x,t\right) = \frac{1}{\sqrt{\pi D_{eff} t}} \exp\left(\frac{x-vt}{4 D_{eff} t}\right) \tag{23}$$

$$H(t,\tau) = e^{-\frac{\alpha}{\beta}(t-\tau)-\alpha\tau} \frac{\tau I_1\left(\frac{2\alpha}{\beta}\sqrt{\beta(t-\tau)\tau}\right)}{\sqrt{\beta(t-\tau)\tau}}$$ (24)

$$\beta = \frac{1-n}{n}\frac{\rho_s c_s}{\rho_f c_f}$$ (25)

Where $PDF_0(x,t)$ represents the probability density function of the residence time without heat transfer between the solid and fluid phase. The parameter $\beta$ (-) represents the ratio between the volume specific heat capacity of the solid phase and the fluid and and $I_1$ is the modified Bessel function of order 1.

[revised manuscript text omitted]

In figure 6 the obtained experimental relationship between Pe and $k_{eff}/k_f$ has been compared with the results obtained by several authors (Levec & Carbonell, 1985; Gunn & Price, 1969; Pfancich, 1963;

Ebach &White, 1958. For the range of Pe investigated the experimental results presents the same order of magnitude of $k_{eff}/k_f$. For low Peclet numbers the experimental value of $k_{eff}/k_f$ is systematically greater than the value of the trend line. This phenomenon can be attributable to the density gradients which altered the flow pattern. Given that the water flows from the bottom to the top and the hot water is fed from the bottom, the buoyancy effect adds to the diffusion effect. This effect seems relevant from low Pe value.

Figure 7 – highlights the experimental correlation between $Nu_v$ and Re. The equation (13) fails to represent the experimental results especially for $M_1$ where they are underestimated of an order of magnitude. For $M_2$ the theoretical model reaches the experimental results having a percentage error of 5 – 35 %.

According to Ando et al. (2013) the volumetric Nusselt number is well represented by equation (14). The exponent $m$ approaches the unit whereas the constant $C$ assumes equal values for $M_1$ and for $M_2$. According to Ando et al. (2013) the coefficient $C$ decreases as the pore diameter (correlated with the particle diameter) increases.

[revised manuscript text omitted]

N. Wakao, S. Kaguei, T. Funazkri, Effects of fluid dispersion coefficients on particle to fluid heat transfer coefficient in packed beds, Chem. Eng. Sci. 34 (1979) 325.

X. Fu, R. Viskanta, J.P. Gore, Measurement and correlation of volumetric heat transfer coefficients of cellular ceramics, Experimental Thermal and Fluid Science 17 (1998) 285±293.

Kouichi Kamiuto T, San San Yee, Heat transfer correlations for open-cellular porous materials.

International Communications in Heat and Mass Transfer 32 (2005) 947–953.

K. Ando1, H. Hirai and Y. Sano. An Accurate Experimental Determination of Interstitial Heat

Transfer Coefficients of Ceramic Foams Using the Single Blow Method. The Open Transport

Phenomena Journal, 2013, 5, 7-12.

[Figure]

**Figure 1. Setup of experimental apparatus**

[Figure]

[Figure]

**Figure 2. Samples of the materials used for the experiments with different average grain sizes $d_p$. a) $d_p$ = 9.2 mm b) $d_p$ = 41.6**
**mm.**

|  | $M_1$ | $M_2$ |
|---|---|---|
| Porosity (-) | 0.47 | 0.53 |
| Average grain size (mm) | 9.21 | 41.65 |
| Average specific surface ($m^{-1}$) | 675.80 | 148.4 |
| Soild density ($Kg \cdot m^{-3}$) | 2210 | 2210 |
| Soil heat capacity ($J \cdot Kg^{-1} \cdot K^{-1}$) | 840 | 840 |
| Soil thermal conductivity ($W \cdot m^{-1} \cdot K^{-1}$) | 2.15 | 2.15 |

**Table 1. Properties of the porous materials**

|  | Re | $q/n \times 10^{-2}$ (m/s) | $v \times 10^{-2}$ (m/s) | $D_{eff} \times 10^{-3}$ ($m^2$/s) | $\alpha$ ($s^{-1}$) | $\beta$ (-) | RMSE | $r^2$ |
|---|---|---|---|---|---|---|---|---|
| $M_1$ | 5.7 | 0.134 | 0.109 | 0.099 | 0.144 |  | 75.659 | 0.9781 |
|  | 9.5 | 0.223 | 0.222 | 0.102 | 0.260 |  | 4.835 | 0.9958 |
|  | 15.5 | 0.361 | 0.321 | 0.116 | 0.403 |  | 1.176 | 0.9984 |
|  | 22.5 | 0.525 | 0.486 | 0.126 | 0.767 | 0.480 | 0.033 | 0.9999 |
| $M_2$ | 23.5 | 0.106 | 0.138 | 0.165 | 0.003 |  | 29.740 | 0.9815 |
|  | 46.9 | 0.211 | 0.248 | 0.273 | 0.006 |  | 7.843 | 0.9886 |
|  | 69.3 | 0.312 | 0.367 | 0.409 | 0.008 |  | 4.742 | 0.9904 |
|  | 105.5 | 0.475 | 0.579 | 0.655 | 0.012 | 0.476 | 1.747 | 0.9944 |

**Table 2. Estimated values of parameters for LTNE model for different Re values.**

|  | Re | Pe | $k_{eff}/k_f$ | Nu | Da |
|---|---|---|---|---|---|
| $M_1$ | 5.7 | 70.05 | 688.83 | 12.35 | 146.28 |
|  | 9.5 | 142.49 | 711.48 | 22.36 | 130.18 |
|  | 15.5 | 205.95 | 811.60 | 34.62 | 139.47 |
|  | 22.5 | 311.64 | 882.13 | 65.91 | 175.46 |
| $M_2$ | 23.5 | 402.32 | 1148.61 | 6.03 | 2.10 |
|  | 46.9 | 721.64 | 1902.80 | 13.42 | 2.61 |
|  | 69.3 | 1068.78 | 2856.49 | 17.85 | 2.34 |
|  | 105.5 | 1684.17 | 4568.91 | 27.07 | 2.25 |

**Table 3. Dimensionless numbers Pe, $k_{eff}/k_f$, Nu and Da calculated for different Re values.**

[Figure]

Figure 3. Temperature distribution for increasing Re values along the porous column filled with material $M_1$. The two curves represent the inlet and the downstream temperature. Squares represent the experimental values, the continuous lines represent the simulated values.

[Figure]

**Figure 4. Temperature distribution for increasing Re values along the porous column filled with material M₂. The two curves represent the inlet and the downstream temperature. Squares represent the experimental values, continuous lines represent the simulated values.**

[Figure]

**Figure 5. Relationship between *Pe* and $k_{eff}/k_f$ .**

 **Figure 6. comparison between the obtained experimental results with other experiments.**

[Figure]

 **Figure 7. Relationship between Pe and Nu$_v$.**

[Figure]

 **Figure 8. Relationship between C×Pe and Nu.**

[Figure]

**Figure 9. heat exchanged $J/(T_{ini} - T_0)$ varying the specific rate $q$ (m/s).**

[Figure]

**Figure 10. Nusselt number – head loss ratio varying Peclet number.**

---

## Author Comment (AC2) · 6 Feb 2018

1) The abstract part should be reduced to highlight the more important contents. The abstract has been reduced by removing the initial part and focusing only on the most important content 2) This paper focuses on the experimental study of forced convection heat transport in porous media. So many related works have been listed in introduction part. Compared to the existing methods and results, what is the main contributions of this paper? The authors should explicitly specify this. The main contributions of the paper have been highlighted by adding this paragraph in the conclusions: 'The main contribution of this study is to investigate on the optimal thermal energy storage of porous materials by analyzing how the grain size and the specific surface affect heat storage properties as well as heat transport in terms of macrodispersion phenomena, heat transfer between solid and fluid phases. This is relevant in order to optimize the efficiency of geothermal installations in aquifers.' 3) Authors claimed that for M2 there is the existence of stagnant zones which reduce the amount of porosity that C1 NPGD Interactive comment Printer-friendly version Discussion paper contributes to fluid flow. Add more analysis for this. More analyses have been inserted by adding the following text: 'In other words, because the coarser material M2 is less well sorted than the less coarse one M1, not all pores of the former are actually interconnected. In geologic materials, based on the connectivity of pores, consequently, the void space can be divided into: interconnected pore, isolated pore, and blind pore (Hu & Huang). Only the pores that are well interconnected provide continuous channels for heat and mass transfer and fluid flow, while the pores that are not part of a continuous channel network do not contribute. These pores are known as noneffective pores, namely, they provide no space for fluid flow and heat transfer in reservoirs.' 4) All the parameters should be explained. Some parts are hard to follow. The unexplained parameters in equations 19), 20) and 21) have been explained. 'Where PDF0(x,t) represents the probability density function of the residence time without heat transfer between the solid and fluid phase. The parameter ïĄć (-) represents the ratio between the volume specific heat capacity of the solid phase and the fluid and and I1 is the modified Bessel function of order 1. .' 5) Please highlight the importance of this experimental study. And clearly indicate what is the new question that is answered in this experimental study? The importance of the experimental study is the investigation on how hydrothermal properties such as grain size and the specific surface affect heat transport in terms of macrodispersion phenomena, heat transfer between solid and fluid phases and heat storage properties. It has been highlighted in text. 6) Add more words for Fig.3 and 4. The caption of Figg 3-4 has been modified into (adding more words): 'Temperature distribution for increasing Re values along the porous column filled with material M1 (M2). The two curves represent the inlet and the downstream temperature. Squares represent the experimental values, the continuous lines represent the simulated values.' 7) Authors indicated that "the behavior of two porous media having different grain sizes and specific surfaces has been observed." If possible, please provide snapshoot for this. A snapshot for the two geological materials having different grain sizes was provided in Figure 2. 'Samples of the materials used for the experiments with different average grain sizes dp. a) dp = 9.2 mm b) dp = 41.6 mm.' 8) What kind of experimental data was acquired? How to interpret these experimental data? All these need to be clearly indicated. The acquired experimental data are: the instantaneous flow rates measured by an ultrasonic velocimeter (DOP3000 by Signal Processing) and the thermal breakthrough curves (BTCs) monitored by the seven thermocouples. The experimental data have been modeled by the analytical solution describing 1D heat transport in semi – infinite domain for instantaneous temperature injection. Moreover the dimensionless numbers Pe, keff/kf, Nu and Da have been evaluated for the different values of Re. In order to highlight the performance of the heat transfer enhancement of the two different porous materials the ratio between the Nusselt number and the hydraulic head loss dh/dx has been calculated as function of the Peclet number.

Please also note the supplement to this comment:
https://www.nonlin-processes-geophys-discuss.net/npg-2017-53/npg-2017-53-AC2-supplement.pdf

**Supplement:**

[revised manuscript text omitted]

Heat transfer dynamics can be represented also by the volumetric Nusselt number $\text{Nu}_v$:

$$\text{Nu}_v = \frac{h s_f d_p^2}{k_f} \qquad (12)$$

Wakao et al. (1979) found the following correlation for the volumetric Nusselt number:

$$\text{Nu}_v = 2.0 + 1.1 \text{Re}^{0.6} \text{Pr}^{1/3} \qquad (13)$$

Fu et al. (1998), Kamiut and Yee (2005) and Ando et al. (2013) based on experimental data found a correlation between the volumetric Nusselt number and Reynolds number:

$$\text{Nu}_v = C \text{Re}^m \qquad (14)$$

The hydrodynamic mixing of the interstitial fluid at the pore scale gives rise to significant thermal dispersion phenomena. Generally, the hydrodynamic mixing is due to the presence of obstruction, flow restriction and turbulent flow. Therefore, the equivalent thermal conductivity in equation (4)

and thermal conductivity in equation (7) is replaced with the effective thermal conductivity $k_{eff}$

which is the sum of thermal conductivity and thermal dispersion conductivity. The effective thermal conductivity depends on various parameters such as mass flow rate, porosity, shape of pores, temperature gradient, and solid and fluid thermal properties (Kaviany, 1995). The following equation can be used to estimate $k_{eff}$.

$$\frac{k_{eff}}{k_f} = \frac{k}{k_f} + K \cdot \text{Pe}^a \qquad (15)$$

Pe represents the Peclet number defined as the product between the Reynolds number Re and

Prandtl number Pr;

$$\text{Pe} = \text{Re} \times \text{Pr} = \frac{\rho_f v d_p}{\mu} \times \frac{c_f \mu}{k_f} = \frac{v d_p}{D_f} \qquad (16)$$

The energy equation representative of the local thermal non equilibrium can be written as:

$$\frac{\partial T_f}{\partial t} = -v \frac{\partial T_f}{\partial x} + D_{eff} \frac{\partial^2 T_f}{\partial x} + \alpha (T_s - T_f) \qquad (17)$$

$$\frac{1-n}{n} \frac{\rho_s c_s}{\rho_f c_f} \frac{\partial T_s}{\partial t} = \frac{1-n}{n} \frac{k_s}{\rho_f c_f} \frac{\partial^2 T_s}{\partial x} - \alpha (T_s - T_f) \qquad (18)$$

With:

$$D_{eff} = \frac{k_{eff}}{\rho_f C_f} \qquad (19)$$

$$\alpha = \frac{h s_f}{n \rho_f C_f} \tag{20}$$

$D_{eff}$ ($L^2 T^{-1}$) is the thermal dispersion and $\alpha$ ($T^{-1}$) is the exchange coefficient.

The thermal dispersion happens due to hydrodynamic mixing of fluid at the pore scale caused by the
nature of the porous medium. Greenkorn (1983) found nine mechanisms responsible of most of the
mixing among which the following: 1) Mixing caused by the tortuosity of the flow channels due to
obstructions: fluid elements starting a given distance from each other and proceeding at the same
velocity will not remain the same distance apart; 2) Existence of autocorrelation in flow paths: in this
case, all pores in a porous medium may not be accessible to a fluid element after it has entered a
particular flow path; 3) Recirculation due to local regions of reduced pressure due to the conversion of
pressure energy into kinetic energy; 4) Hydrodynamic dispersion in a capillary caused by the velocity
profile produced by the adhering of the fluid to the wall; 5) Molecular diffusion into dead-end pores: as
solute rich front passes the pore. After the front passes, the solute will diffuse back out and thus,
dispersing.

Using the analogy with the solute transport the Damköhler number Da (Leij et al., 2012) can be
introduced in order to evaluate the influence of heat transfer between the fluid and solid phases on
the convection phenomena:

$$\mathrm{Da} = \frac{\alpha L}{v} \tag{21}$$

When Da reaches the unit the heat transfer time scale is comparable with the convection time scale
and the LTNE exists between solid and fluid phases. At very high values of Da the heat transfer
time scale is much lower than convective time scale and the LTE condition exists between solid and
fluid phases. Finally, at very low values of Da the heat transfer phenomena between solide and fluid
phase can be neglected.

Neglecting the first term on the right side of the Equation 18, the analytical solution of the system
equations describing 1D heat transport in semi – infinite domain for instantaneous temperature
injection is given by Goltz and Roberts (1986). According to this analytical solution, the probability
of density function $PDF_{LTNE}$ of the residence time for LTNE condition can be written as:

$$PDF_{LTNE}\left(x,t\right) = e^{\alpha t} PDF_0\left(x,t\right) + \alpha \int_0^t H\left(t,\tau\right) PDF_0\left(x,t\right) d\tau \tag{22}$$

With:

$$PDF_0\left(x,t\right) = \frac{1}{\sqrt{\pi D_{eff} t}} \exp\left(\frac{x-vt}{4 D_{eff} t}\right) \tag{23}$$

$$H(t,\tau) = e^{-\frac{\alpha}{\beta}(t-\tau)-\alpha\tau} \frac{\tau I_1\left(\frac{2\alpha}{\beta}\sqrt{\beta(t-\tau)\tau}\right)}{\sqrt{\beta(t-\tau)\tau}}$$ (24)

$$\beta = \frac{1-n}{n}\frac{\rho_s c_s}{\rho_f c_f}$$ (25)

Where $PDF_0(x,t)$ represents the probability density function of the residence time without heat transfer between the solid and fluid phase. The parameter $\beta$ (-) represents the ratio between the volume specific heat capacity of the solid phase and the fluid and and $I_1$ is the modified Bessel function of order 1.

[revised manuscript text omitted]

In figure 6 the obtained experimental relationship between Pe and $k_{eff}/k_f$ has been compared with the results obtained by several authors (Levec & Carbonell, 1985; Gunn & Price, 1969; Pfancich, 1963;

Ebach &White, 1958. For the range of Pe investigated the experimental results presents the same order of magnitude of $k_{eff}/k_f$. For low Peclet numbers the experimental value of $k_{eff}/k_f$ is systematically greater than the value of the trend line. This phenomenon can be attributable to the density gradients which altered the flow pattern. Given that the water flows from the bottom to the top and the hot water is fed from the bottom, the buoyancy effect adds to the diffusion effect. This effect seems relevant from low Pe value.

Figure 7 – highlights the experimental correlation between $Nu_v$ and Re. The equation (13) fails to represent the experimental results especially for $M_1$ where they are underestimated of an order of magnitude. For $M_2$ the theoretical model reaches the experimental results having a percentage error of 5 – 35 %.

According to Ando et al. (2013) the volumetric Nusselt number is well represented by equation (14). The exponent $m$ approaches the unit whereas the constant $C$ assumes equal values for $M_1$ and for $M_2$. According to Ando et al. (2013) the coefficient $C$ decreases as the pore diameter (correlated with the particle diameter) increases.

[revised manuscript text omitted]

N. Wakao, S. Kaguei, T. Funazkri, Effects of fluid dispersion coefficients on particle to fluid heat transfer coefficient in packed beds, Chem. Eng. Sci. 34 (1979) 325.

X. Fu, R. Viskanta, J.P. Gore, Measurement and correlation of volumetric heat transfer coefficients of cellular ceramics, Experimental Thermal and Fluid Science 17 (1998) 285±293.

Kouichi Kamiuto T, San San Yee, Heat transfer correlations for open-cellular porous materials.

International Communications in Heat and Mass Transfer 32 (2005) 947–953.

K. Ando1, H. Hirai and Y. Sano. An Accurate Experimental Determination of Interstitial Heat

Transfer Coefficients of Ceramic Foams Using the Single Blow Method. The Open Transport

Phenomena Journal, 2013, 5, 7-12.

[Figure]

**Figure 1. Setup of experimental apparatus**

[Figure]

[Figure]

**Figure 2. Samples of the materials used for the experiments with different average grain sizes $d_p$. a) $d_p$ = 9.2 mm b) $d_p$ = 41.6**
**mm.**

|  | $M_1$ | $M_2$ |
|---|---|---|
| Porosity (-) | 0.47 | 0.53 |
| Average grain size (mm) | 9.21 | 41.65 |
| Average specific surface ($m^{-1}$) | 675.80 | 148.4 |
| Soild density ($Kg \cdot m^{-3}$) | 2210 | 2210 |
| Soil heat capacity ($J \cdot Kg^{-1} \cdot K^{-1}$) | 840 | 840 |
| Soil thermal conductivity ($W \cdot m^{-1} \cdot K^{-1}$) | 2.15 | 2.15 |

**Table 1. Properties of the porous materials**

|  | Re | $q/n \times 10^{-2}$ (m/s) | $v \times 10^{-2}$ (m/s) | $D_{eff} \times 10^{-3}$ ($m^2$/s) | $\alpha$ ($s^{-1}$) | $\beta$ (-) | RMSE | $r^2$ |
|---|---|---|---|---|---|---|---|---|
| $M_1$ | 5.7 | 0.134 | 0.109 | 0.099 | 0.144 |  | 75.659 | 0.9781 |
|  | 9.5 | 0.223 | 0.222 | 0.102 | 0.260 |  | 4.835 | 0.9958 |
|  | 15.5 | 0.361 | 0.321 | 0.116 | 0.403 |  | 1.176 | 0.9984 |
|  | 22.5 | 0.525 | 0.486 | 0.126 | 0.767 | 0.480 | 0.033 | 0.9999 |
| $M_2$ | 23.5 | 0.106 | 0.138 | 0.165 | 0.003 |  | 29.740 | 0.9815 |
|  | 46.9 | 0.211 | 0.248 | 0.273 | 0.006 |  | 7.843 | 0.9886 |
|  | 69.3 | 0.312 | 0.367 | 0.409 | 0.008 |  | 4.742 | 0.9904 |
|  | 105.5 | 0.475 | 0.579 | 0.655 | 0.012 | 0.476 | 1.747 | 0.9944 |

**Table 2. Estimated values of parameters for LTNE model for different Re values.**

|  | Re | Pe | $k_{eff}/k_f$ | Nu | Da |
|---|---|---|---|---|---|
| $M_1$ | 5.7 | 70.05 | 688.83 | 12.35 | 146.28 |
|  | 9.5 | 142.49 | 711.48 | 22.36 | 130.18 |
|  | 15.5 | 205.95 | 811.60 | 34.62 | 139.47 |
|  | 22.5 | 311.64 | 882.13 | 65.91 | 175.46 |
| $M_2$ | 23.5 | 402.32 | 1148.61 | 6.03 | 2.10 |
|  | 46.9 | 721.64 | 1902.80 | 13.42 | 2.61 |
|  | 69.3 | 1068.78 | 2856.49 | 17.85 | 2.34 |
|  | 105.5 | 1684.17 | 4568.91 | 27.07 | 2.25 |

**Table 3. Dimensionless numbers Pe, $k_{eff}/k_f$, Nu and Da calculated for different Re values.**

[Figure]

Figure 3. Temperature distribution for increasing Re values along the porous column filled with material $M_1$. The two curves represent the inlet and the downstream temperature. Squares represent the experimental values, the continuous lines represent the simulated values.

[Figure]

**Figure 4. Temperature distribution for increasing Re values along the porous column filled with material M₂. The two curves represent the inlet and the downstream temperature. Squares represent the experimental values, continuous lines represent the simulated values.**

[Figure]

**Figure 5. Relationship between *Pe* and $k_{eff}/k_f$ .**

 **Figure 6. comparison between the obtained experimental results with other experiments.**

[Figure]

 **Figure 7. Relationship between Pe and Nu$_v$.**

[Figure]

 **Figure 8. Relationship between C×Pe and Nu.**

[Figure]

**Figure 9. heat exchanged $J/(T_{ini} - T_0)$ varying the specific rate $q$ (m/s).**

[Figure]

**Figure 10. Nusselt number – head loss ratio varying Peclet number.**